# Humanization of Pan-HLA-DR mAb 44H10 Hinges on Critical Residues in the Antibody Framework

**DOI:** 10.3390/antib13030057

**Published:** 2024-07-16

**Authors:** Audrey Kassardjian, Danton Ivanochko, Brian Barber, Arif Jetha, Jean-Philippe Julien

**Affiliations:** 1Program in Molecular Medicine, The Hospital for Sick Children Research Institute, 686 Bay Street, Toronto, ON M5G 0A4, Canada; 2Department of Immunology, University of Toronto, 1 King’s College Circle, Toronto, ON M5S 1A8, Canada; 3Department of Biochemistry, University of Toronto, 1 King’s College Circle, Toronto, ON M5S 1A8, Canada

**Keywords:** MHC class II targeting, antibody humanization, framework regions, Vernier Zone

## Abstract

Reducing the immunogenicity of animal-derived monoclonal antibodies (mAbs) for use in humans is critical to maximize therapeutic effectiveness and preclude potential adverse events. While traditional humanization methods have primarily focused on grafting antibody Complementarity-Determining Regions (CDRs) on homologous human antibody scaffolds, framework regions can also play essential roles in antigen binding. Here, we describe the humanization of the pan-HLA-DR mAb 44H10, a murine antibody displaying significant involvement of the framework region in antigen binding. Using a structure-guided approach, we identify and restore framework residues that directly interact with the antigen or indirectly modulate antigen binding by shaping the antibody paratope and engineer a humanized antibody with affinity, biophysical profile, and molecular binding basis comparable to that of the parental 44H10 mAb. As a humanized molecule, this antibody holds promise as a scaffold for the development of MHC class II-targeting therapeutics and vaccines.

## 1. Introduction

Monoclonal antibodies (mAbs) are an increasingly important class of therapeutics with uses across diverse indications, forecasted to constitute a ~USD 500 billion industry by 2030 [1]. Currently approved mAbs in clinical use can be categorized based on their “human-ness”–the degree to which their constant and variable region sequences resemble those found in the human antibody repertoire. Of these, antibodies with higher proportions of non-human content display a greater propensity for eliciting anti-drug antibody (ADA) responses that often result in rapid in vivo clearance, leading to suboptimal exposure and loss of efficacy [2,3,4,5]. Additionally, highly immunogenic antibodies can lead to unwanted adverse effects [6], precluding the use of such therapeutics in clinical settings. As such, humanization constitutes a crucial step in antibody development to lower the immunogenicity of animal-derived antibodies for therapeutic use in humans.

As antibody Complementarity-Determining Regions (CDRs) mediate the majority of contacts with the antigen, early humanization methods involved transplanting the CDRs from an antibody of foreign origin onto human antibody framework regions [7]. This strategy, termed CDR grafting, remains highly used today as a cornerstone of antibody humanization. Several factors should be considered when using CDR grafting. First, there exist multiple different amino acid numbering schemes and CDR definitions with varying lengths, each associated with its benefits and drawbacks for grafting [8,9]. To minimize the number of non-human residues in the humanized molecule, using shorter CDR definitions may be advantageous. However, the selective grafting of as few parental residues as possible can come at the expense of antigen specificity and affinity.

Another consideration for CDR grafting is the choice of the human framework as an acceptor scaffold. Here, approaches that maximize either sequence or structural homology to the parental framework have been proposed to increase the likelihood of positive outcomes since homologous frameworks provide the same scaffold for the CDRs implicated in antigen binding [8]. In addition to the selection of appropriate CDRs and framework regions, back-mutations of select residues in the framework are often necessary, as they are known to play a role in shaping antibody paratopes and may also directly contact the antigen [10,11,12,13,14]. Thus, striking a balance between maximizing human content whilst retaining the parental antibody’s specificity and affinity for the antigen of interest can be a tricky endeavor.

Though there are currently no approved antibody therapeutics targeting MHC class II, anti-class II antibodies have been investigated for a variety of applications, such as modulating graft rejection [15,16], preventing and treating Experimental Autoimmune Encephalomyelitis (EAE) [17], and most prominently as cancer immunotherapies and vaccines [18,19,20,21]. These antibodies have been proposed to act via various mechanisms of action, exerting their functions by directly stimulating immune effector cells or blocking inhibitory immune interactions. The antibody of interest in this report, mAb 44H10, was isolated from BALB/c mice immunized with cells from a non-T, non-B acute lymphoblastic leukemia (ALL) cell line and further discovered to react to MHC class II molecules by radioimmunoprecipitation and tissue distribution studies [22]. Murine mAb 44H10 was shown to target an epitope on the HLA-DR molecule conserved in humans [23,24], suggesting its potential clinical benefit across populations irrespective of HLA haplotype. Previously, this mAb was used as the scaffold for immunotargeting vaccines delivering protein antigens to MHC class II-expressing antigen-presenting cells [23,25], successfully inducing potent adjuvant-free antigen-specific antibody responses. As a highly modular platform with several cargo anchoring points, mAb 44H10 could be employed for the delivery of diverse antigens, with potential applications in the development of both vaccines and therapeutics. Thus, the humanization of mAb 44H10 would be highly beneficial in advancing such technologies toward pre-clinical development and potential eventual use in humans.

Here, we describe the structure-guided humanization of mAb 44H10, leveraging CDR grafting methodologies further guided by critical insights derived from the structural characterization of both the parental antibody and intermediate humanized candidates. The final engineered humanized antibody, built on a clinically de-risked framework scaffold, retained the antigen-binding, biophysical properties, and molecular basis of HLA-DR reactivity of the parental 44H10 mAb.

## 2. Results

### 2.1. Humanization of mAb 44H10 by CDR Grafting Is Insufficient to Retain Affinity to HLA-DR

To generate a humanized candidate that retained the specificity and affinity of parental 44H10 for HLA-DR, we first employed a conventional CDR grafting strategy using the IMGT scheme to define CDRs to maximize antibody “human-ness” (Figure 1A). Three human heavy chains (*IGHV4-59*, *IGHV2-26* and *IGHV4-4*) and three human light chains (*IGKV1-16*, *IGKV1-39* and *IGKV1-6*) variable framework regions were selected as scaffolds for grafting based on sequence homology to the parental murine antibody (*IGHV2-9*, *IGKV9-124*) using the IMGT/DomainGapAlign database [26,27], resulting in nine candidates (versions 1–9; V1–9) generated through combinatorial pairing in the first round of humanization (Figure 1B, Appendix A). These candidates were screened for binding to the HLA-DR-expressing B-lymphoblastoid cell line BJAB [28] by flow cytometry (Figure 1C). As none of the nine initial candidates displayed significant reactivity to the BJAB cells, an additional nine candidates were generated by grafting the slightly longer Kabat-defined 44H10 CDRs onto previously selected frameworks in a second round of humanization (V10–V18) (Figure 1A,B, Appendix A). While none of these additional candidates displayed reactivity to BJAB cells comparable to parental 44H10, one candidate (V17) displayed superior binding relative to the others (Figure 1D). The binding of V17 to BJAB cells was further characterized using a full antibody titration, where a greater than 50-fold difference in binding EC_50_ between parental 44H10 and V17 was measured (Figure 1E). In biolayer interferometry (BLI) binding assays, parental 44H10 displayed an apparent affinity of 1.5 nM for recombinant HLA-DR, whereas V17 did not produce a measurable binding signal at the concentrations tested (Figure 1F, Appendix A). Thus, conventional CDR grafting methods using two distinct CDR definitions generated a humanized candidate with only modest HLA-DR reactivity, displaying reduced affinity to the antigen relative to parental 44H10. Furthermore, the nonuniform binding of the various humanized candidates generated in the first two rounds of humanization suggested that the largely conserved CDRs grafted were not solely responsible for antigen binding, implicating the divergent human framework regions in binding specificity and affinity.

### 2.2. Critical mAb 44H10 Light Chain Framework Residues Mediate Contacts with HLA-DR

To understand the additional determinants of binding affinity, the presence of paratope framework residues was examined in a crystal structure of the parental 44H10 Fab bound to HLA-DR (PDB: 8EUQ). Analysis of the buried surface area (BSA) revealed that residues outside Kabat-defined CDRs contributed ~44% of the total interface BSA, primarily attributed to residues in the light chain framework region 3 (LFR3). In particular, K60 and R66 in this region contributed the light chain’s greatest per-residue BSAs (Figure 2A) and mediated several direct contacts with the HLA-DR heterodimer. The L-K60 side chain formed a salt bridge with E65 on the HLA-DR β chain and contributed van der Waals (vdW) contacts with several surrounding residues on both antigen chains (Figure 2B). L-R66 formed a hydrogen bond with the P81 main chain on the HLA-DR α chain and additional vdW interactions with surrounding α chain residues (Figure 2B). L-K60 and L-R66 found in parental 44H10 were changed to serine and glycine in V17, respectively (Figure 2A). The crystal structure of the unliganded V17 Fab solved to a 1.95 Å resolution (Appendix A) highlighted how the shorter side chains could not replicate these binding interactions mediated by the parental mAb to HLA-DR (Figure 2B).

Given the observed involvement of L-K60 and L-R66 for antigen binding, we generated V17-based humanized candidates with single or double back-mutations in these two light chain framework residues (V19–21). All three candidates with back-mutations displayed a ~3-fold EC_50_ improvement relative to V17 (Figure 2C), supporting the contributory role of these residues in antigen binding. Unlike V17, V21 (bearing both S60K and G66R light chain back-mutations) displayed measurable binding to recombinant HLA-DR in BLI experiments (Figure 2D, Appendix A). Together, these data demonstrate the direct implication of specific 44H10 framework residues in antigen binding. Nonetheless, the reversion of L-K60 and L-R66 did not fully compensate for the decreased antigen binding affinity observed between parental 44H10 and V17.

### 2.3. Framework Residues in the 44H10 Heavy Chain Modulate Antibody Paratope Stability and Antigen Binding

Given that the reversion of framework residues making direct antigen contacts did not enhance the HLA-DR binding affinity of our humanized candidate to a level comparable to parental 44H10, we hypothesized that antibody framework regions additionally impacted antigen binding through indirect mechanisms. As such, we next examined non-binding candidates generated in the initial humanization efforts to investigate why certain selected frameworks were not suitable for antigen binding. While V17 exhibited binding to BJAB cells, we did not detect binding by V14, although both mAbs shared the exact same light chains (Figure 1D, Appendix A). To investigate this observed discrepancy in binding, we sought to gain structural insights into how the heavy chain frameworks used in V14 (*IGHV4-59*) and V17/V21 (*IGHV2-26*) functioned as the primary discriminants of antigen binding. To this end, the crystal structures of the unliganded V14 and V21 Fabs were solved to 2.5 and 1.7 Å resolutions, respectively (Appendix A).

The crystal structure of V21 Fab revealed high structural homology to parental 44H10, displaying similar loop conformations in antigen-interfacing areas and minimal root-mean-square deviations (RMSD) between the two structures (Figure 3A,B). Conversely, the overlay of the V14 Fab crystal structure with that of parental 44H10 revealed greater deviations in the conformation of select loops, with the HCDR3 displaying RMSDs up to 8 Å relative to the parental structure (Figure 3A,B). Given that the HCDR3 sequences in parental, V14, and V21 Fabs were identical, we hypothesized that differences in HCDR3 loop conformations observed in the crystal structures of these Fabs could be due to differences in support provided by the surrounding framework. Closer inspection of the structures revealed a slight but significant shift in the heavy chain framework region 1 (HFR1) of V14 relative to parental/V21 Fabs that could conceivably impact the directly adjacent HCDR3 loop (Figure 3C). This difference was primarily mediated by the HFR3 residue at position 71, where K71 (as found in parental/V21 Fabs) formed a hydrogen bond with the main chain of L29 in the adjacent HFR1 loop (Figure 3C and Appendix A). On the other hand, a valine at this same position in V14 was unable to mediate the same HFR1-stabilizing contact with L29, resulting in a distinct shift in the HFR1 loop. This effect was compounded by the HFR3 residue found at position 78, where a small valine residue in parental/V21 Fabs enabled the K71-L29 interaction. In contrast, a bulky phenylalanine found in V14 created steric hindrance preventing this HFR1-stabilizing contact (Figure 3C).

To further investigate the effect of heavy chain residues 71 and 78 on antigen binding, we generated a V14 variant Fab with single and double V71K/F78V mutations and, inversely, a V21 Fab variant with single and double K71V/V78F mutations. Unlike wild-type V14, which expectedly did not bind recombinant HLA-DR in BLI experiments, the V14-V71K/F78V double mutant generated a measurable binding signal with a K_D_ of 206 nM (Figure 3D, Appendix A). Conversely, the opposite mutations in V21 completely abolished the binding of the wild-type antibody to HLA-DR (Figure 3G, Appendix A). As expected, a lysine at position 71 paired with a phenylalanine at position 78 in the context of either V14 or V21 Fabs enabled low-level antigen binding, and this binding was enhanced when K71 was instead paired with a valine (Appendix A). No binding was observed for any Fab lacking a lysine at position 71, irrespective of the residue at position 78. Differential scanning calorimetry (DSC) further investigated the effect of these mutations, where V71K/F78V mutations increased V14 Fab’s melting temperature by 1.6 °C, suggesting increased thermostability (Figure 3E). Conversely, K71V/V78F mutations decreased V21 Fab’s melting temperature by 2.6 °C, indicating decreased thermostability (Figure 3H).

To quantify the influence of residues at heavy chain positions 71 and 78 at atomic level, 200 ns molecular dynamics (MD) simulations were performed for the unliganded wild-type and double mutant V14 and V21 Fabs. Per-residue root mean square fluctuations (RMSFs) were calculated for each variable domain; in the case of V14, the V71K/F78V double mutation exhibited decreased RMSF values, primarily within the HFR1, HCDR2, and HCDR3, suggestive of lower flexibility and increased paratope stability (Figure 3F). Conversely, K71V/V78F mutations in V21 increased the V21 heavy chain RMSFs (mainly in the HFR1, HCDR2, and HFR3) throughout the simulations, suggesting increased heavy chain flexibility and lower paratope stability (Figure 3I). Together, these data demonstrate the pivotal role of HFR3 residues 71 and 78 in enabling the binding of parental and humanized 44H10 candidates to HLA-DR and in modulating antibody paratope flexibility and thermostability.

### 2.4. Humanized mAb 44H10 Displays Favorable Biophysical Properties

Having elucidated the direct and indirect roles of the antibody framework on antigen binding, and prioritizing clinically de-risked antibody frameworks with favorable developability properties, we generated one final humanized 44H10 candidate by pairing an *IGHV4-59* heavy chain (as in V14) and *IGHK1-16* light chain (as in V17/21), grafted with Kabat 44H10 CDRs and bearing back-mutations in key heavy chain (H-K71, H-V78) and light chain (L-K60, L-R66) framework residues. The IgG format of this new lead candidate, named V22, displayed improved binding relative to all previous versions in both BLI and flow cytometry binding assays (Figure 4A,B, Appendix A), with a less than two-fold reduction in EC_50_ relative to parental 44H10 IgG measured by flow cytometry.

To assess the relative thermostability of parental 44H10 and V22, purified IgGs of both antibodies were stored at either −20 °C, 4 °C or 40 °C and evaluated weekly in a panel of biophysical assays for up to 4 weeks. Throughout the study and at all temperatures tested, both antibodies remained consistent in size and polydispersity relative to baseline measurements (Figure 4C and Appendix A) and maintained consistent HLA-DR-binding properties (Figure 4D,E, Appendix A). Importantly, across all assays, V22 displayed similar biophysical and antigen-binding properties as parental 44H10. Thus, structure-guided framework mutations paired with conventional CDR grafting methods successfully generated a humanized 44H10 antibody candidate built on a clinically de-risked framework region displaying comparable biophysical properties as parental 44H10.

### 2.5. Parental and Humanized 44H10 Fabs Target HLA-DR in Highly Analogous Binding Modes

To compare the molecular basis of the HLA-DR reactivity of our lead humanized candidate to that of the parental 44H10 antibody, the crystal structure of the V22 Fab-HLA-DR complex was solved to a 3.1 Å resolution (Appendix A). As expected, V22 targeted the same epitope on HLA-DR as parental 44H10 (Figure 5A), with similar BSA contributions to the antigen interface for both antibodies (Figure 5B). Furthermore, key interactions mediated by the antibody CDRs were highly conserved between the parental and humanized Fabs (Figure 5C–E and Appendix A). In the HCDR3, H-Y100B and H-Y96 mediated hydrogen bonds with the main chains of P87 and V89 in the HLA-DR α chain, respectively, as seen in the parental 44H10 Fab-HLA-DR crystal structure (Figure 5C and Appendix A). In the LCDR1, L-Y32 formed a hydrogen bond with the main chain of α-N84 (Figure 5D and Appendix A). Finally, key hydrogen bonds were formed by L-Y49, L-S52 and L-T53 in the antibody LCDR2, contacting HLA-DR residues β-Q63, β-E64 and α-D142 (Figure 5E and Appendix A). Due to weak electron density surrounding the L-K60 side chain in this co-crystal condition, we unfortunately could not observe contacts mediated by this key FW residue in the V22-HLA-DR crystal structure. Furthermore, ion coordination by the LFR3 loop in the V22 structure, which was not observed in the parental structure solved from a different co-crystallization condition, did not permit the direct comparison of contacts mediated by L-R66. Nonetheless, the structural elucidation of the V22 Fab-HLA-DR complex demonstrates that the humanized 44H10 antibody, V22, targets HLA-DR in a binding mode comparable to that of parental 44H10.

## 3. Discussion

The success of humanization by conventional CDR grafting is highly variable, often requiring time- and resource-intensive rounds of optimization. Here, the humanization of mAb 44H10 was initiated by rounds of CDR grafting and was ultimately made possible by insights gained from the co-crystal structure of the parental 44H10 Fab with HLA-DR. Successful engineering was further guided by the structures of intermediate candidates generated throughout the humanization process.

Both traditional and emerging humanization methods emphasize the conservation of CDRs because they are known to mediate the majority of contacts with the antigen [7]. However, the antibody framework can also have important roles in antigen binding. Indeed, about 20% of antigen-binding residues, on average, fall outside the CDRs, and these are suggested to be at least as critical to antigen binding as residues within CDRs [14,29]. Here, the crystal structure of the 44H10 Fab in complex with HLA-DR revealed a substantial contribution of the light chain framework to antigen binding, and indicated an important involvement of residues L-K60 and L-R66. Supporting binding data showed affinity improvements when these residues were back-mutated in our initial humanized candidate generated by CDR-grafting. These data support the use of structure-guided design paired with conventional CDR grafting methods to identify residues outside the CDRs implicated in antigen binding, and the targeting of these residues by reverse mutagenesis to improve the affinity of humanized candidates.

Framework regions can also indirectly modulate antibody paratopes by providing support critical for shaping CDRs. Such framework residues constitute the “Vernier Zone” (VZ), and have been shown to impact antigen binding, antibody energetics, and antibody activity [10,11,12,13]. Notably, one of the most recognized examples of a VZ residue is position 71 in the antibody heavy chain (H71), which was also identified here as critical for enabling humanized 44H10 binding to HLA-DR. This finding is supported by previous reports demonstrating the role of H71 in defining HCDR1/2 conformations in several distinct antibodies [30,31,32,33]. In addition to its effect on antigen binding, H71 also affected the paratope stability of the humanized candidates. This correlation between high-affinity interactions and decreased protein flexibility is expected, given that a rigid structure can more tightly bind to a single epitope, a phenomenon that has been widely studied in the context of affinity maturation [34,35,36,37,38,39]. In the humanization of 44H10, the sole reversion of H71 from a valine to a lysine did not restore antigen binding in a non-binding candidate (V14) due to the presence of a bulky neighboring phenylalanine at position H78, sterically inhibiting the K71-L29 interaction determined to stabilize the CDRs. Concurrent reversion of residues at H71 and H78 was required to enable higher affinity antigen binding, highlighting the importance of considering residue pairs in conformational environments. In several other reports, amino acid substitutions at H71 did not impact CDR conformations and antigen binding [40,41]. Thus, systematic grafting of H71 or other VZ residues may not be necessary in all cases, and one must consider the importance of each VZ position in the context of given CDRs and antibody-antigen interactions. Altogether, our data augment the findings in previous reports on the impact of non-antigen-interfacing antibody framework regions on stabilizing CDRs and shaping paratopes, further supporting the use of structural studies to identify critical VZ residues and supplement CDR-grafting strategies for antibody humanization. Additional humanization methods such as superhumanization [42,43] and resurfacing [44] also rely on the availability of structural information to guide humanization efforts. More recent approaches aim to harness artificial intelligence algorithms for humanization via in silico modeling [45,46,47,48]. While these certainly hold potential, many of these also rely on the availability of structural information to increase the probability of success.

Beyond antigen affinity, the use of monoclonal antibodies in humans requires optimizing several of their biophysical attributes for ease of manufacturing and distribution [49]. As such, certain antibody framework regions have biophysical properties better suited for clinical development and are preferred scaffolds for humanization. In our lead humanized antibody (V22), an *IGHV4-59* heavy chain with engineered framework mutations was prioritized over the *IGHV2-26* heavy chain used in the alternative lead (V21) due to its preferred biophysical properties [50] and prevalence in clinical-stage mAbs [51]. Indeed, Fabs using either an *IGHV2-26* heavy chain or an engineered *IGHV4-59* heavy chain (with FW back-mutations) paired with nearly identical light chains (differing only by two amino acids) had melting temperatures differing by ~5.5 °C (Figure 3E,H), illustrating the superior thermostability of the *IGHV4-59* framework. Another consideration is the frequency of chosen framework regions in various human populations, as V and J genes are polymorphic [52,53,54], and the use of rare V alleles in antibody therapeutics has been associated with a higher risk of immunogenicity [55]. Thus, commonly used V alleles serve as a good starting point for the engineering of humanized antibodies as they are likely to be recognized evenly in all human populations.

We previously used mAb 44H10 as the scaffold of an immunotargeting vaccine (ITV) against SARS-CoV-2, whereby the SARS-CoV-2 Spike protein RBD was fused to a chimeric mouse-human 44H10 IgG, effectively targeting this antigen to MHC class II-expressing antigen-presenting cells [23]. Here, the humanization of the 44H10 antibody benefits the development of the ITV platform for potential use in humans by minimizing the risk of immunogenicity of the antibody scaffold and focusing the response on the immunotargeted cargo antigen.

### Limitations of the Study

X-ray crystallography enables the visualization of antibodies in a static crystalline state, and, therefore, may underestimate the flexibility of loops. As such, interpretation of the crystal structures presented here, especially those at lower resolutions, should be performed with this limitation in mind, and are best done supplemented with the molecular dynamics data also provided in this report.

## 4. Methods

### 4.1. Mammalian Cell Lines and Culture Conditions

Female mammalian cells (FreeStyle^TM^ 293-F cells, Thermo Fisher Scientific, Waltham, MA, USA; HEK 293S, GnT I^−/−^ cells, ATCC, Manassas, VA, USA) were cultured in suspension in GIBCO^TM^ FreeStyle^TM^ 293 Expression Medium (Thermo Fisher Scientific) at 37 °C in a Multitron Pro Shaker (Infors HT, Anjou, QC, Canada) with 70% humidity, 8% CO_2_ and rotating at 130 rpm. B lymphoblastoid BJAB cells (DSMZ, Braunschweig, Germany) [28] were grown in RPMI medium supplemented with 10% fetal bovine serum (FBS) and 1% penicillin/streptomycin (all from Thermo Fisher Scientific) in a 37 °C, 5% CO_2_ incubator.

### 4.2. Analysis of mAb 44H10 CDRs and Framework Regions

The heavy chain and light chain sequences of the murine antibody 44H10 were analyzed using the abYsis server [56] for the identification of IMGT and Kabat CDRs. Human framework regions with high sequence homology to parental 44H10 were identified using the IMGT/DomainGapAlign database [26,27].

### 4.3. Plasmid Design and Synthesis

DNA plasmids for the expression of all proteins described in this work were designed in the pcDNA3.4 TOPO mammalian expression vector and optimized for *Homo sapiens* expression at GeneArt (Invitrogen, Waltham, MA, USA). These constructs were amplified in DH5α competent *E. coli* cells and purified using PureLink HiPure Plasmid Maxiprep Kits (Invitrogen).

### 4.4. Expression and Purification of Recombinant Humanized 44H10 IgGs

FreeStyle 293-F cells were split to a density of 0.8 × 10^6^ cells/mL at least one hour before transfection. Cells were transfected using PEI MAX^®^ (Polysciences, Warrington, PA, USA) following manufacturer instructions at a 1:5 DNA to PEI MAX^®^ ratio. 90 µg of plasmid DNA was used for transfection (2:1 ratio of heavy and light chain DNA plasmids) for every 200 mL of cell culture. Transfected cells were incubated in a 37 °C, 5% CO_2_ shaking incubator for 5 to 7 days to allow for the expression and pairing of heavy and light chain gene products. Transfected cell culture supernatants were collected and filtered through 0.22 µM Steritop filters (Millipore Sigma, Burlington, MA, USA) before loading onto protein A affinity columns using the ÄKTA start protein purification system (Cytiva Life Sciences, Marlborough, MA, USA). Samples were eluted with 100 mM glycine, pH 2.2 and immediately neutralized with 1 M Tris, pH 9. Elution fractions were concentrated using Amicon 30K Ultra-0.5 mL Centrifugal Filters (Millipore Sigma) and further purified by size exclusion chromatography on a Superdex 200 Increase 10/300 GL (Cytiva Life Sciences) in PBS (Thermo Fisher Scientific).

### 4.5. Expression and Purification of Recombinant Humanized 44H10 Fabs

Fab heavy and light chain plasmids were transfected into FreeStyle 293-F cells as described above. Recombinant Fabs were purified by KappaSelect affinity chromatography with 100 mM glycine, pH 2.2 elution and immediate 1 M Tris, pH 9 neutralization, followed by MonoS ion exchange chromatography using 20 mM NaOAc, pH 5.6 ± 1M KCl, and size exclusion chromatography on a Superdex 200 Increase 10/300 GL in 20 mM Tris, pH 8, 150 mM NaCl (TBS) (Cytiva).

### 4.6. Expression and Purification of Recombinant HLA-DR

HLA-DR α and β chain plasmids (previously described [23]) were co-transfected in a 1:1 ratio (50 µg total per 200 mL of cell culture) into HEK 293S (GnT I^−/−^) cells. Recombinant HLA-DR was purified by affinity chromatography via a HisTrap Ni-NTA column (Cytiva) and eluted using 20 mM Tris, pH 8.0, 500 mM imidazole. Subsequent size exclusion chromatography was performed in TBS, pH 8 on a Superdex 200 Increase 10/300 GL (Cytiva). For crystallization trials, purified HLA-DR was additionally subjected to overnight treatment with EndoH and TEV proteases at ratios of 5:1 and 20:1, respectively, for deglycosylation and cleavage of the Fos/Jun zipper and His-tags. A second round of affinity and size exclusion purifications was performed for HLA-DR before complexation with V22 Fab for crystallization trials.

### 4.7. Detection of Binding to BJAB cells by Flow Cytometry

BJAB cells were collected in a conical tube and centrifuged at 300× *g* for 5 min. Cell pellets were resuspended in staining buffer (PBS, 2% FBS, 0.05% NaN_3_) at 1 × 10^6^ cells/mL, and 200 µL of the cell suspension was dispensed into the wells of a polystyrene, V-bottom 96-well plate (Greiner Bio-One, Monroe, NC, USA) for staining. Cells were centrifuged at 300× *g* for 5 min, and then incubated in Fc Block (BD Biosciences, Mississauga, ON, Canada) for 10 min at room temperature. Purified humanized 44H10 IgGs were then added to the cells and incubated for 1 h at 4 °C. Cells were washed twice, then stained with AF488 AffiniPure Goat Anti-Human IgG, Fcγ fragment specific (1:1000, Jackson ImmunoResearch, West Grove, PA, USA) for 30 min at 4 °C. After two additional washes, cells were resuspended in propidium iodide (1:100, Thermo Scientific) to exclude dead cells and debris. Samples were acquired on BD LSR II or LSR Fortessa cell analyzers using the BD FACSDiva™ Software v9.0, and further analyzed using the FlowJo™ Software v10.

### 4.8. Biolayer Interferometry for Measurement of HLA-DR Binding to Humanized 44H10 mAbs

Real-time analysis of binding kinetics was measured using the Octet RED96 BLI system (Sartorius, Goettingen, Germany). Baseline, association, and dissociation steps were conducted at 25 °C for 180 s in kinetics buffer (PBS, pH 7.4, 0.01% BSA, 0.002% Tween). For the detection of IgG binding to HLA-DR, recombinant HLA-DR (uncleaved) was loaded onto Penta-His Biosensors (FortéBio, Fremont, CA, USA) at 10 µg/mL until a threshold response of 0.7 nm, and association events were measured by dipping loaded biosensors into wells containing a two-fold serial dilution of 44H10, V17, V21 or V22 IgGs at a 125 nM starting concentration. For the detection of Fab binding to HLA-DR, purified wild-type or mutant V14 and V21 Fabs were loaded onto FAB2G biosensors (FortéBio) at 10 µg/mL until a threshold response of 0.7 nm, and association events were measured by dipping loaded biosensors into wells containing a two-fold serial dilution of TEV-cleaved HLA-DR at a 500 nM starting concentration. For the accelerated thermostability assays, purified 44H10 or V22 IgGs were loaded onto Protein A biosensors (FortéBio) at 10 µg/mL until a threshold response of 0.7 nm, and association events were measured by dipping loaded biosensors into wells containing a two-fold serial dilution of TEV-cleaved HLA-DR at a 500 nM starting concentration. Dissociation was measured by transfer of biosensors back into buffer-containing wells. Biosensors were regenerated in 100 mM glycine, pH 1.5 (Penta-His) or pH 2.2 (FAB2G and Protein A). Kinetics data were analyzed using the FortéBio Octet Data Analysis software 9.0.0.6, and curves were fitted to a 1:1 binding model for the calculation of K_D_, k_on_ and k_off_.

### 4.9. Differential Scanning Calorimetry

Melting temperatures (T_m_) for V14 and V21 wild-type and mutant Fabs were measured on a NanoDSC (TA Instruments, New Castle, DE, USA). T_m_s were obtained by measuring changes in the partial molar heat capacity of proteins at constant pressure during a temperature ramp from 20 to 95 °C with 1 °C increments. All measurements were done at a protein concentration of 0.5 mg/mL in PBS. T_m_s were determined by the NanoAnalyze software by fitting the data to the Gaussian with T_onset_ model.

### 4.10. Molecular Dynamics (MD)

Double swap-mutations of heavy chain residues 71 and 78 were generated with COOT (Version 0.9.6) [57] using the coordinates from the three-dimensional X-ray crystal structures of Fab V14 and V21. The Simple Mutate tool was used to generate models of Fab V14-V71K-F78V and Fab V21-K71V-V78F. Rotamers for each swap mutant residue were manually selected to recapitulate the rotamers observed in the other structure. MD simulations were computed using the Desmond module available in BioLuminate (Schrödinger Release 2022-4) [58]. The coordinates of Fabs V14, V21, V14-V71K-F78V, and V21-K71V-V78F were first pre-processed using the Protein Preparation Wizard using the default settings with the following additions: crystallization solvent components were removed, the C-termini were capped, and hydrogen atoms were incorporated at pH 7.4 with PROPKA where only the orientations of the hydrogen atoms were optimized with the OPLS4 force field. Additionally, all waters with fewer than 1 hydrogen bond to a non-water group were deleted. The structures of each Fab were positioned within orthorhombic boxes buffered by distances of 10.0 Å × 10.0 Å × 10.0 Å beyond the protein and then solvated using the OPLS4 force field with explicit single point charge (SPC) water molecules using the System Builder module. All solvated protein systems were neutralized by the addition of Cl^−^ counter ions in a final salt concentration of 0.15 M NaCl. Prior to the production run, all solvated systems were subjected to Desmond’s default energy minimization protocol. The production simulations were performed using the OPLS4 force field for a duration of 200 ns. The NPT ensemble class was used with a temperature of 310.15 K and a pressure of 1.01325 bar. A recording interval of 200.0 ps was employed generating approximately 1000 frames per production run. The calculations of RMSF values at each residue were performed using the MDAnalysis library (Version 2.5.0) [59] in Python (Version 3.7.16).

### 4.11. Dynamic Light Scattering

Dynamic light scattering (DLS) analysis was performed using a DynaPro Plate Reader III (Wyatt Technology). 20 μL of each protein at 1 mg/mL were added to a 384-well black, clear bottom plate (Corning, Corning, NY, USA) and measured at a fixed temperature of 25 °C with a duration of 5 s per read. Particle hydrodynamic radii (Rh) and polydispersity (% Pd) were obtained from the accumulation of ten reads from duplicate samples using the Dynamics software (Wyatt Technology, Goleta, CA, USA).

### 4.12. Crystallization of Humanized 44H10 Fabs

V17 Fab in TBS pH 8 was concentrated to 10.4 mg/mL and mixed with a crystallization buffer of 20% PEG3350, 0.2 M potassium nitrate pH 6.9, as well as crystal seeds previously obtained in a condition of 0.1 M HEPES pH 7.5, 25% PEG3350. Crystals appeared after 1 day and grew steadily for 28 days, at which time they were cryoprotected in 15% (*v*/*v*) ethylene glycol before being flash-frozen in liquid nitrogen. V14 Fab in TBS pH 8 was concentrated to 9.7 mg/mL and mixed with a crystallization buffer of 0.17 M Sodium acetate, 25.5% (*w*/*v*) PEG4000, 15% (*v*/*v*) glycerol, 0.085 M Tris pH 8.5, as well as crystal seeds previously obtained in a condition of 20% glycerol, 0.16 M magnesium chloride, 24% PEG4000, 0.08 M Tris pH 8.5. Crystals appeared after 1 day and grew steadily for 11 days, at which time they were cryoprotected in 15% (*v*/*v*) ethylene glycol before being flash-frozen in liquid nitrogen. V21 Fab in TBS pH 8 was concentrated to 10 mg/mL and mixed with a crystallization buffer of 20% PEG3350 and 0.2M potassium iodide, as well as crystal seeds previously obtained in a condition of 0.1 M HEPES pH 7.5, 25% PEG3350. Crystals appeared after 1 day and grew steadily for 18 days before being flash-frozen in liquid nitrogen. All seeded crystallization trays were set with a ratio of 2:1:3 (protein:seed:crystallization buffer).

### 4.13. Co-Crystallization of the V22 Fab-HLA-DR Complex

V22 Fab was mixed with recombinant HLA-DR in a 1.5-molar excess, and excess Fab was removed via size exclusion chromatography (Superdex 200 Increase 10/300 GL, Cytiva) in TBS pH 8. The protein complex was concentrated to 10 mg/mL and mixed with a crystallization buffer of 20% (*w*/*v*) PEG 6000, 0.1 M citric acid, pH 4.0 as well as crystal seeds previously obtained in a condition of 19% (*v*/*v*) Isopropanol, 19% (*w*/*v*) PEG4000, 5% (*v*/*v*) glycerol and 0.095 M sodium citrate pH 5.6 with a ratio of 2:1:3 (protein:seed:crystallization buffer). Crystals appeared after 48 days and grew steadily until day 128, at which time they were cryoprotected in 15% (*v*/*v*) glycerol before being flash-frozen in liquid nitrogen.

### 4.14. X-ray Diffraction Data Collection, Processing and Refinement

X-ray diffraction data were collected at the 23-ID-B or 23-ID-D beamlines at the Advanced Photon Source (APS) or the CMCF-ID-D beamline at the Canadian Light Source (CLS). All unliganded Fab datasets were processed, merged and scaled using XDS, XSCALE and Xprep [60]. The V22 Fab-HLA-DR complex dataset was processed using the Staraniso server [61] with a diffraction-limit surface (*I*_mean_/σ(*I*_mean_)) of 1.2. Structures were determined by molecular replacement using Phaser [62]. Structure refinements were performed using phenix.refine [63] and Coot [57]. Access to all software was supported through SBGrid [64]. Fab-antigen interactions were analyzed using the PDBePisa server for BSA analysis [65]. Protein-Ligand Interaction Profiler (PLIP) [66] and CCP4 Contacts [67] were used for detection of hydrogen bonds, hydrophobic interactions, salt bridges and vdW contacts. PyMOL (Version 2.5.5, Schrödinger, LLC, New York, NY, USA) was used for RMSD calculations using the RmsdByResidue script.

## Figures and Tables

**Figure 1 antibodies-13-00057-f001:**
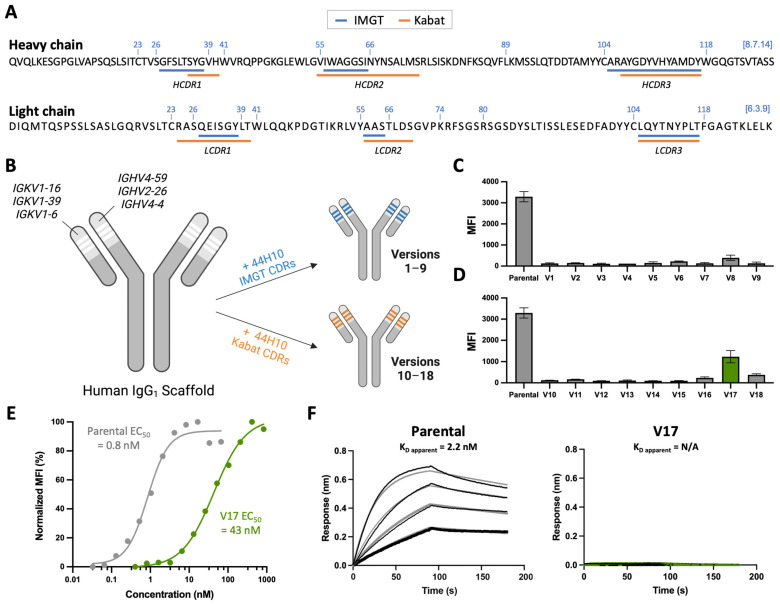
Humanization by CDR grafting is insufficient to retain affinity to HLA-DR. (**A**) Parental 44H10 heavy and light chain amino acid sequences annotated with IMGT (blue) and Kabat (orange) CDR definitions. The IMGT unique numbering of key conserved amino acids is provided in blue above each sequence. (**B**) Conventional CDR-grafting process used to generate humanized 44H10 versions 1–18, grafting IMGT- or Kabat-defined CDRs onto selected human antibody frameworks. (**C**,**D**) Flow cytometry screening of humanized 44H10 V1–9 (**C**) and V10–18 (**D**) binding to BJAB cells (10 µg/mL concentration). The candidate showing modest antigen binding (V17) is colored in green. (**E**) Flow cytometry titration curves of parental 44H10 and V17 IgGs binding to BJAB cells. (**F**) BLI binding profile of parental 44H10 and V17 IgGs to recombinant HLA-DR, where black lines represent measured binding and gray/green curves correspond to the data fitted to a 1:1 binding model.

**Figure 2 antibodies-13-00057-f002:**
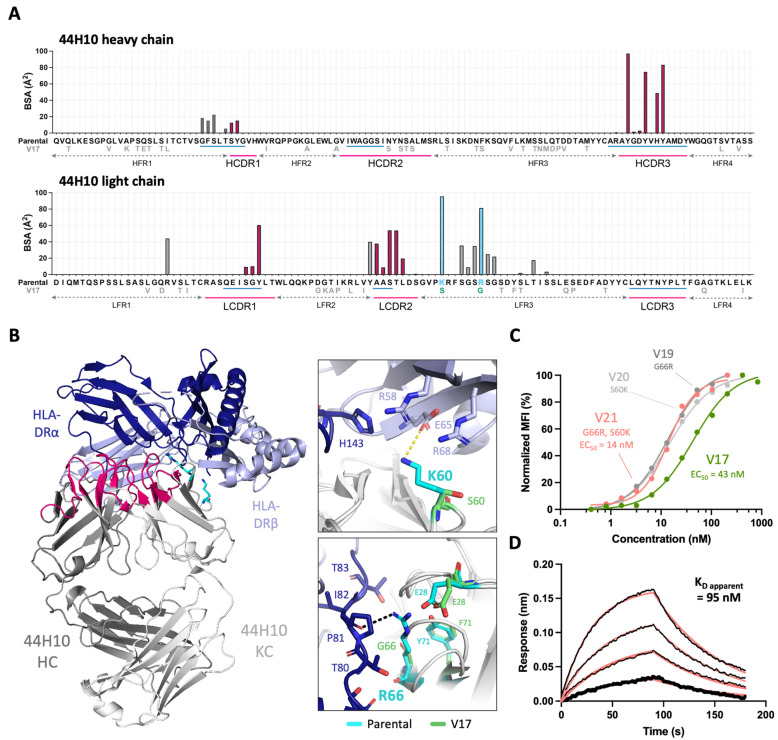
Key 44H10 light chain framework residues mediate contacts with HLA-DR and improve the affinity of humanized candidates. (**A**) Buried surface area (BSA) contribution of parental 44H10 CDR (Kabat, pink underlines) and framework residues contacting HLA-DR as determined by PDBePISA interface analysis (PDB: 8EUQ). Corresponding IMGT-defined CDRs are annotated by blue underlines. Amino acid deviations from the parental sequence found in V17 are indicated below the parental sequence. BSA contributed by CDR residues are colored in pink, and framework residues L-K60 and L-R66 contributing significant BSA are colored in light blue. (**B**) Cartoon representation of the parental 44H10 Fab-HLA-DR complex, with 44H10 CDRs colored in hot pink and light chain framework residues L-K60 and L-R66 shown as cyan sticks. Insets are close-up views of key interactions mediated by L-K60 and L-R66 in parental 44H10, which cannot be mediated by L-S60 and L-G66 in V17 (represented as green sticks). A black dotted line indicates a hydrogen bond, and a yellow dotted line indicates a salt bridge. (**C**) Flow cytometry titration curves comparing V17, V19 (G66R back-mutation), V20 (S60K back-mutation) and V21 (S60K and G66R back-mutations) IgGs binding to BJAB cells. (**D**) BLI binding profile of V21 IgG to recombinant HLA-DR, where black lines represent measured binding and salmon curves correspond to the data fitted to a 1:1 binding model.

**Figure 3 antibodies-13-00057-f003:**
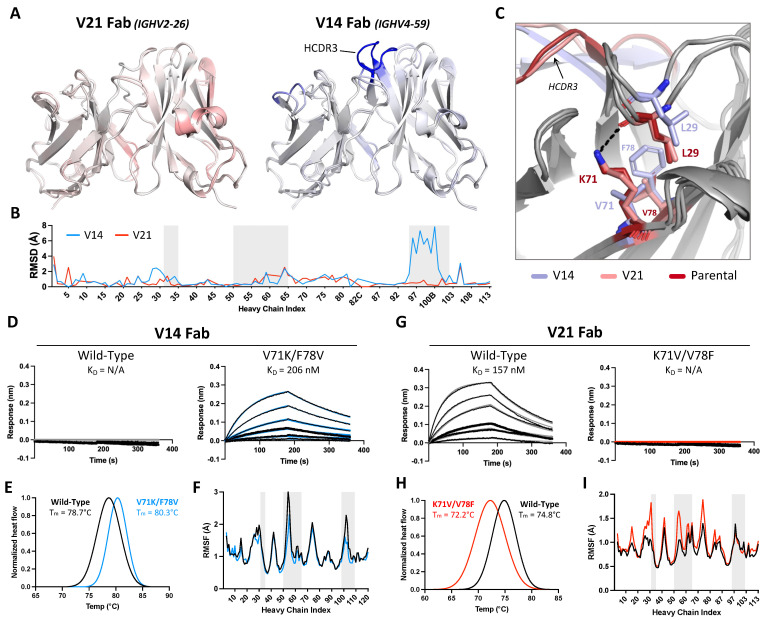
Framework residues in the 44H10 heavy chain modulate antibody paratope stability and antigen binding. (**A**) Overlay of V21 (left) and V14 (right) Fab variable region crystal structures with parental 44H10 colored by RMSD, where increasing saturation indicates greater deviation from the parental structure. (**B**) Quantification of V14 and V21 heavy chain variable regions RMSDs relative to parental 44H10. Areas with gray shading indicate Kabat HCDRs. (**C**) Close-up view of stabilizing interactions mediated by residues H-K71 and H-V78 with H-L29 in parental 44H10 (red) and V21 (salmon), which cannot be mediated by H-V71 and H-F78 in V14 (light blue). A black dotted line indicates a hydrogen bond. (**D**–**I**) Biophysical characterization of wild-type V14 Fab and mutant V14 Fab with V71K/F78V heavy chain mutations (**D**–**F**) and wild-type V21 Fab and mutant V21 Fab with K71V/V78F heavy chain mutations (**G**–**I**). (**D**,**G**) BLI binding profiles of V14 and V21 wild-type and mutant Fabs to recombinant HLA-DR, where black lines represent measured binding and colored curves correspond to the data fitted to a 1:1 binding model. (**E**,**H**) Differential Scanning Calorimetry (DSC) profiles of V14 and V21 wild-type and mutant Fabs with calculated melting temperatures (T_m_). (**F**,**I**) RMSF plots derived from MD simulations of V14 and V21 wild-type and mutant Fabs indicating changes in heavy chain variable region flexibility. Areas with gray shading represent Kabat HCDRs.

**Figure 4 antibodies-13-00057-f004:**
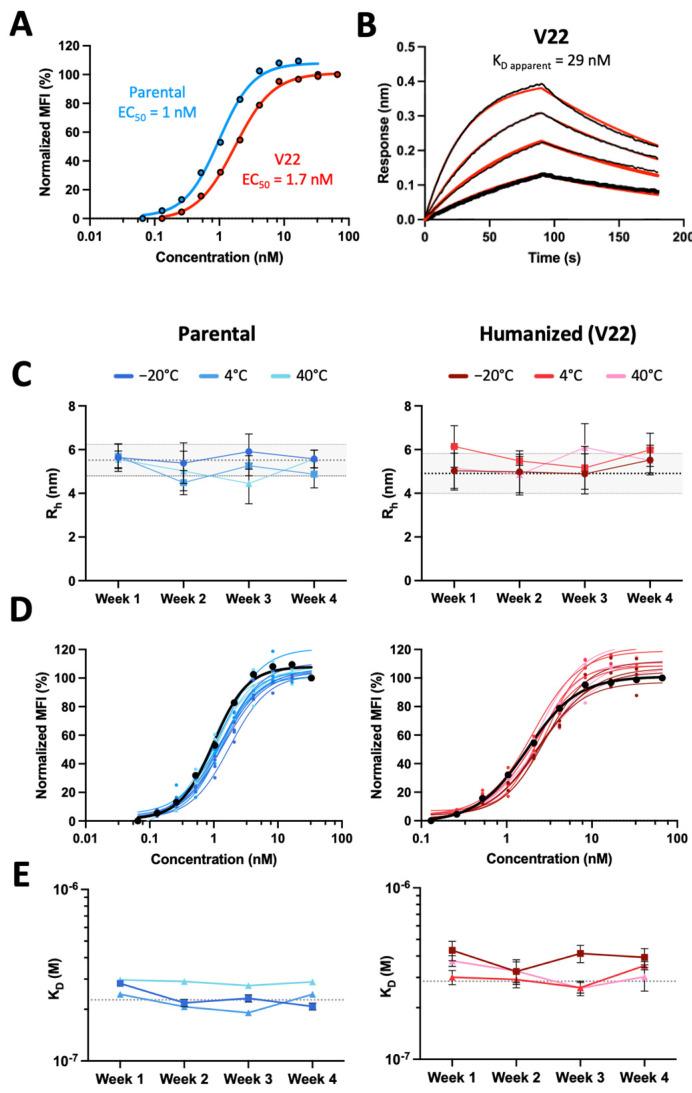
Humanized 44H10 antibody displays biophysical properties and thermostability similar to the parental 44H10 antibody. (**A**) Flow cytometry titration curves comparing parental and V22 IgG binding to BJAB cells. (**B**) BLI binding profiles of V22 IgG to HLA-DR, where black lines represent measured binding and red curves correspond to the data fitted to a 1:1 binding model. (**C**–**E**) Biophysical characterization of parental 44H10 and V22 IgGs kept at various storage temperatures for up to four weeks. (**C**) Hydrodynamic radii (R_h_) of 44H10 and V22 IgGs measured by DLS, with error bars representing the polydispersity of each measurement. Baseline Rh and polydispersity for each molecule are indicated by a dotted line and shaded area. (**D**) Binding of 44H10 and V22 IgGs to BJAB cells measured by flow cytometry. Binding of each molecule measured at baseline is shown as a black curve. (**E**) K_D_ affinities of 44H10 and V22 IgGs binding to recombinant HLA-DR, measured by BLI. The baseline K_D_ for each molecule is indicated by a dotted line.

**Figure 5 antibodies-13-00057-f005:**
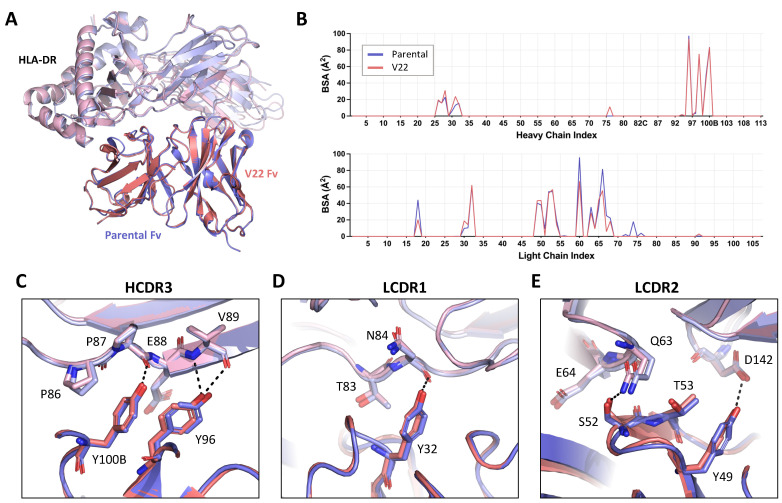
Lead humanized 44H10 candidate V22 binds HLA-DR comparably to parental 44H10. (**A**) Overlay of parental 44H10 Fab-HLA-DR (PDB: 8EUQ) and V22 Fab-HLA-DR crystal structures aligned in PyMOL. Parental and V22 Fabs are shown in blue and salmon, respectively, with HLA-DR from each structure in a lighter shade of the Fab’s color. (**B**) Buried surface area (BSA) contribution of 44H10 and V22 heavy and light chain residues contacting HLA-DR as determined by PDBePISA interface analysis. (**C**–**E**) Close-up views of key antigen contacts mediated by the V22 (salmon) HCDR3 (**C**), LCDR1 (**D**) and LCDR2 (**E**), replicating those observed in the parental 44H10 Fab-HLA-DR structure (blue). Black dotted lines indicate hydrogen bonds.

## Data Availability

The crystal structures reported in this work have been deposited to the Protein Data Bank and are publicly available as of the date of publication. Accession numbers are listed as follows: 9B74, 9B75, 9B76, 9B7B.

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
