# Peer review of "Humanization of Pan-HLA-DR mAb 44H10 Hinges on Critical Residues in the Antibody Framework"

_2073-4468, 2024, doi:10.3390/antib13030057_

Round 1
Reviewer 1 Report
Comments and Suggestions for Authors
Humanization of animal-derived monoclonal antibodies is a common practice in the pharmaceutical industry to reduce the risk of eliciting an anti-drug antibody response. In this study, Kassardjian A et al. aimed to humanize the mouse monoclonal antibody 44H10, which targets human HLA-DR and holds various clinical applications ranging from protecting against graft rejection to treating certain autoimmune diseases, cancer therapy, and vaccines.
Beginning with CDR grafting, they demonstrated that this commonly used, simple strategy is insufficient to retain antibody affinity to the HLA-DR target. They then performed step-by-step structure-guided back mutations in combination with CDR grafting. Their results clearly demonstrated that both the light chain and heavy chain framework residues significantly contribute to target binding and the thermodynamic stability of this specific antibody. The light chain framework residues L-K60 and L-R66 mediate direct contact with HLA-DR, and the heavy chain residues H-K71 and H-V78 indirectly modulate antibody paratope stability and antigen binding.
Their final humanized product, V22, which incorporates all these back mutations and CDR grafting, demonstrated very similar binding structures, as evidenced by the co-crystal structure of the humanized Fab with HLA-DR, and the same thermodynamic properties as the parental 44H10.
This study is well-designed, the methods and results are solid, and the data are clearly presented. The manuscript is well-written. The successful humanization of 44H10 provides a typical example of using structure-guided framework mutations paired with conventional CDR grafting methods for future humanization practices. I strongly recommend accepting the manuscript without major revision.
Author Response
This study is well-designed, the methods and results are solid, and the data are clearly presented. The manuscript is well-written. The successful humanization of 44H10 provides a typical example of using structure-guided framework mutations paired with conventional CDR grafting methods for future humanization practices. I strongly recommend accepting the manuscript without major revision.
We thank the Reviewer for the positive assessment of our work.
Reviewer 2 Report
Comments and Suggestions for Authors
Julien and coworkers describe a tour de force through the humanization of an mouse antibody 44H19 that is particularly interesting for therapeutic use since it broadly addresses HLA-DR. The authors could show through a set of X-ray structure analyses of the wildtype antibody and of various humanized versions that direct interaction of framework residues significantly contributes to target protein binding. The final candidate, for which also a X-ray structure exists still displays reduced affinity (double digit nM in BLI compared to single digit for the wildtype clone) but displays rather good binding on cells. The paper is very well written and certainly merits publication in the journal Antibodies since this finding and strategy might also be of value for other humanization campaigns.
Minor points:
Fig. 1E,2C, 4A: It would be nice to see expression of Ec50 as nM rather than microgram/ml since this can be considered as apparent KD.
Li 194: In contract should read in contrast
The authors performed DLS as a method for determining the monodispersity of the antibody. In addition, it would be nice to see the SEC (size exclusion chromatography) profile of the antibody after purification
Author Response
The paper is very well written and certainly merits publication in the journal Antibodies since this finding and strategy might also be of value for other humanization campaigns.
We thank the Reviewer for the positive assessment of our work and its potential impact.
Fig. 1E,2C, 4A: It would be nice to see expression of Ec50 as nM rather than microgram/ml since this can be considered as apparent KD.
Figures 1E, 2C and 4A have now been updated to express the EC50 in nM.
Li 194: In contract should read in contrast
We thank the Reviewer for noting this typo. It has now been corrected.
The authors performed DLS as a method for determining the monodispersity of the antibody. In addition, it would be nice to see the SEC (size exclusion chromatography) profile of the antibody after purification.
As requested, we have now added Figure S2, which shows the SEC profiles of the parental and V22 IgGs.